# Lead in Air, Soil, and Blood: Pb Poisoning in a Changing World

**DOI:** 10.3390/ijerph19159500

**Published:** 2022-08-02

**Authors:** Howard W. Mielke, Christopher R. Gonzales, Eric T. Powell, Sara Perl Egendorf

**Affiliations:** 1Department of Pharmacology, Tulane School of Medicine, New Orleans, LA 70112, USA; 2Lead Lab, Inc., New Orleans, LA 70119, USA; chrisgc99@gmail.com (C.R.G.); powellet2@gmail.com (E.T.P.); 3Cornell Atkinson Center for Sustainability, Ithaca, NY 14853, USA; spe34@cornell.edu

**Keywords:** atmospheric Pb, urban soil Pb, spatiotemporal soil Pb and blood Pb, primary Pb prevention

## Abstract

(1) Background: Leaded petrol became a worldwide vehicle fuel during the 20th century. While leaded petrol was totally banned on 30 August 2021, its lead (Pb) dust legacy remains in the environment as soil Pb. The health impacts of Pb are well known and risks occur when exposures are above zero. The inextricable links between air Pb, soil Pb, and blood Pb are not widely A. Exposure risks continue even after banning leaded petrol and must be explored. (2) Methods: This article evaluates selected examples of temporal measurements of atmospheric Pb and human Pb exposure and the effect of soil Pb on blood Pb. Several search engines were used to find articles on temporal changes in air Pb and human Pb exposures. New Orleans studies provided empirical data on the association between soil Pb and blood Pb. (3) Results: Vehicle Pb emission trends are closely associated with air Pb and blood Pb. Air Pb deposited in soil becomes a reservoir of Pb dust that is known to be remobilized into the atmosphere. (4) Conclusions: The dust from leaded petrol continues to pose major exposure risks to humans. Exogenous sources of Pb in soil and its remobilization into air along with endogenous bone Pb establish the baseline exposure of children and adults. Reducing human exposure to Pb requires novel policies to decrease exogenous contact from the reservoir of Pb in soil and curtailing remobilization of soil Pb into the atmosphere. Mitigating exposure to soil Pb must therefore play a central role in advancing primary prevention.

## 1. Introduction

*“Now I feel fairly sure of what I am going to say, and that is this: … If the case involved ingested lead and if [the physician] had a good medical education and if he knew the man had a stomachache and certain other symptoms of lead poisoning, he would size it up as lead poisoning. But if the material is inhaled—and its symptomology is altered when it is inhaled, because of its wider distribution in the body—it is extremely likely that nine hundred and ninety-nine ordinary physicians out of a thousand would fail to recognize the condition as lead poisoning.”* Yandel Henderson, 1925 [1].

This study addresses integrating atmospheric lead (Pb), soil Pb, and blood Pb for understanding spatiotemporal dynamics of Pb poisoning, especially in urban settings. The 1925 Yandell Henderson warning about the effects of inhaled air Pb on physiology and human organ systems suggested that Pb aerosols are especially important and underrecognized [1]. Current studies relating air Pb and blood Pb support Professor Henderson’s early concerns.

A major issue for the research community was the inability to measure clinically, environmentally, and appropriately small quantities of Pb. This capability only became widely available in the late 1960s and early 1970s [2]. A major application was the measurement of blood Pb and conducting clinical studies. The combination of blood Pb and human clinical studies revealed that Pb exposure was excessive, and the public health response was to lower the blood Pb guidelines [2].

The use of leaded petrol was approved despite Henderson’s concerns about inhalation. The lead industry promoted leaded petrol and within a few decades, Pb contamination became an international public health disaster [3]. The blood Pb of children and adults fluctuates depending on the culture and regulations in each country regarding the use of leaded petrol. Children are the most susceptible group to Pb exposure. The consequences of children’s Pb exposure includes damaging effects on intelligence and behavior and negative effects on personality traits of the adult population [4,5]. By 30 August 2021, nearly a century after Henderson’s statements, leaded petrol for highway use was banned throughout the world [6].

Lead-based paints, with high Pb content, have often been assumed to be the major source of lead poisoning. The Lead Industry Association (LIA) actively dissuaded any notion that sources of Pb beyond lead-based paint have been involved in lead poisoning. On 14 April 1969, the board of directors of the LIA stated, “*It should be a primary objective of any LIA program, or LIA participation in other programs, aimed at resolving the childhood lead poisoning problem to keep attention focused on old, leaded paint as its primary source and to make clear that other sources of lead are not significantly involved* [7].” The critical issue is the failure to include Pb aerosols from Pb additives in petrol as a major source of Pb exposure, a bias that continues [8]. As Figure 1 illustrates, the amount of Pb used in the U.S. as Pb additives in petrol (gasoline) was about equal to the amount of Pb in lead-based paint [9]. 

The focus on measuring blood Pb and ascribing the exposure to lead-based paint missed obvious environmental signals such as atmospheric Pb and settled Pb dust concentrations [10]. Eventually, the same analytical instruments were adopted to measure air Pb and soil Pb [11], but the findings were often ignored [12]. The purpose of this article is to conduct an integrative review to (1) demonstrate the links between air Pb, soil Pb, and blood Pb, and (2) identify mitigation strategies that limit multiple Pb dust sources and decrease human Pb exposure.

## 2. Materials and Methods

Several search engines, including Google Scholar, Scopus, and Web of Science, were used to obtain articles and open literature on temporal changes of air Pb, soil Pb, and human Pb exposures [13]. We conducted an integrative review to provide an overview of the knowledge base and re-conceptualize the links between air Pb, soil Pb, and blood Pb [14]. Google Scholar was of particular importance for identifying journal titles and authors connected with subjects, obtaining literature such as conference proceedings and articles that would not ordinarily be included in other indexing services, locating obscure references that are difficult to find in conventional databases, and locating information on incomplete citations [13]. We applied the following key words and Boolean terms: blood Pb OR blood lead; air Pb OR atmospheric Pb; leaded gas OR petrol Pb; and we joined each of these in searches with AND. Applying these key words to each search engine uncovered a selection of relevant articles listed in Table 1.

## 3. Results

Temporal changes recorded in air Pb and blood Pb are highlighted in selected Figure 2, Figure 3, Figure 4 and Figure 5 in Section 3.1. In addition, Table 1 lists studies from the United Kingdom, New Zealand, Mexico, and Durban, South Africa [20,21,22,23], which show similar responses between atmospheric Pb and blood Pb as observed in Figure 2, Figure 3, Figure 4 and Figure 5. Finally, two reviews on declines in air Pb and blood Pb demonstrate the links between exogenous (outside the body) and endogenous (inside the body) declines. One review is from Korea [24] and the second review is from northwestern Europe [25]. In Section 3.2, data-dependent studies for London [26] and New Orleans [27] provide insight into spatiotemporal changes in blood Pb and soil Pb. These studies demonstrate that air Pb from leaded petrol plays a key role in exogenous interactions between air Pb, soil Pb, and human Pb poisoning.

### 3.1. Blood Pb Results from Sweden, China, Germany, and Australia

#### 3.1.1. Sweden, Yearly Measurements of Blood Pb in Swedish Children

From 1978 to 1994, in small Swedish cities there was a gradual reduction of blood Pb coinciding with a gradual reduction in leaded petrol [15].Sweden’s petrol became Pb-free in 1995.In 1995–2007, children’s blood Pb declined and stabilized at ~2 µg/d (Figure 2).

**Figure 2 ijerph-19-09500-f002:**
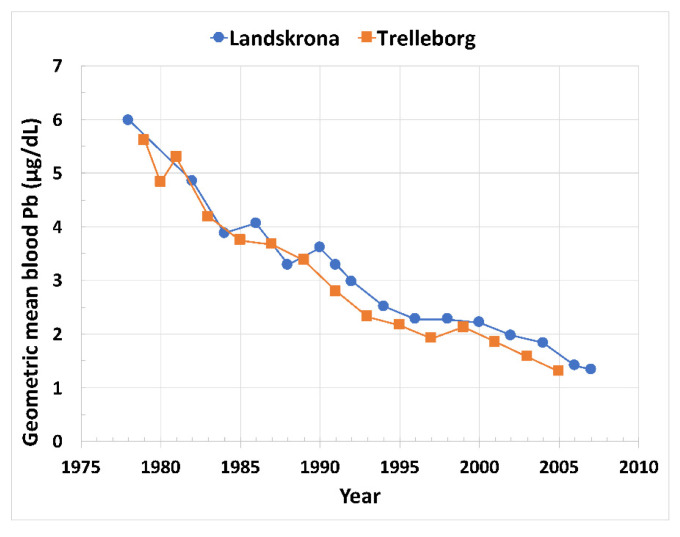
Blood Pb decreases in two small Swedish cities, Landskrona and Trelleborg. This graph was modified from the published data using WebPlotDigitizer Version 4.5 (Pacifica, CA, USA) and Microsoft Excel Version 16.54 (Microsoft Corporation Redmond, Washington, DC, USA).

#### 3.1.2. China, Asir Pb and Blood Pb of Urban and Rural Children

This study noted three phases in blood Pb change [16].In 1997–2000, before the phasedown of leaded petrol, blood Pb increased rapidly.After the phasedown, in 2001–2013, blood Pb decreased rapidly.According to the raw data, in 2014–2015, blood Pb stabilized.The raw data blood Pb are from urban surveys, while air Pb is from multiple industrial surveys. As a result, the two surveys are not necessarily associated.

**Figure 3 ijerph-19-09500-f003:**
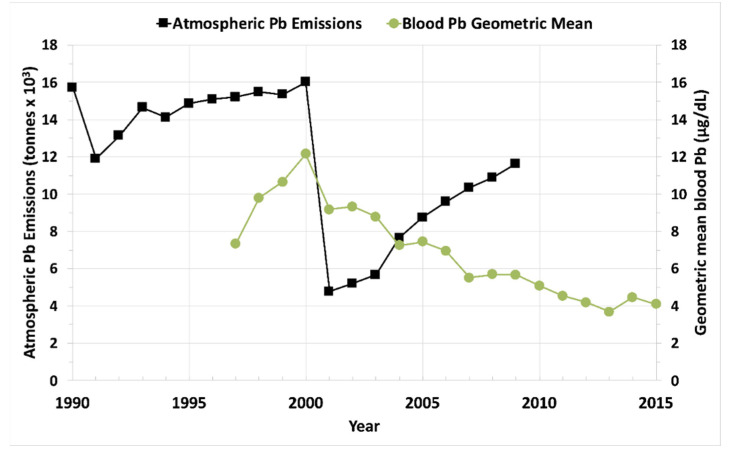
Based on the raw data provided in atmospheric Pb and blood Pb in urban and rural communities appear to be out of sync. The main interest is 2000 when China banned leaded petrol. Air Pb dropped precipitously followed by a sharp, although not as steep, decline in blood Pb, a subsequent more gradual decrease, and a flattened response after 2014.

#### 3.1.3. Germany, Pb Exposure >35 Years of 20–29-Year-Old Compared with the Same Aged US young Adults

Data from the early 1980s to 2019 reveal a blood Pb decrease of about 87% [17].The US 20–29-year-old young adults represent rural, suburban, and urban areas.The trends in human exposure closely correlate with air Pb levels.Since 2010, blood Pb in German young adults have flattened to ~1 µg/dL.

**Figure 4 ijerph-19-09500-f004:**
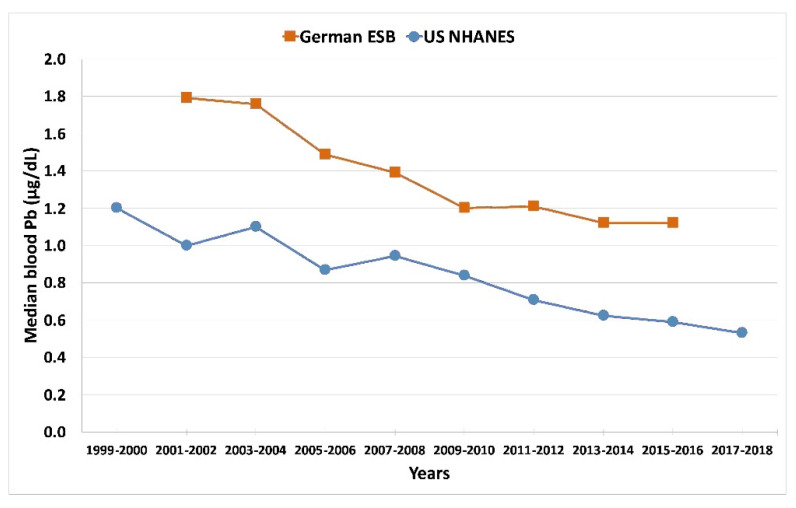
Blood Pb data for 20–29-year-old urban German population compared to the US NHANES (nationally represented data) for the same age group. Compiled from Table 3 in ref. [17] and from NHANES data [18]. The graph was created using Microsoft Excel Version 16.54 (Redmond, WA, USA).

#### 3.1.4. Australia, the Rise, Fall, and Remobilization of Industrial Lead

Australian environmental Pb emissions began in the 1880s.Environmental archives contain multiple natural Pb isotopic values.The decline in leaded petrol coincides with urban blood Pb.Leaded petrol continued to decline but urban blood Pb stabilized at <2 µg/dLAround mines and smelters, blood, air, and soil Pb remain elevated.

**Figure 5 ijerph-19-09500-f005:**
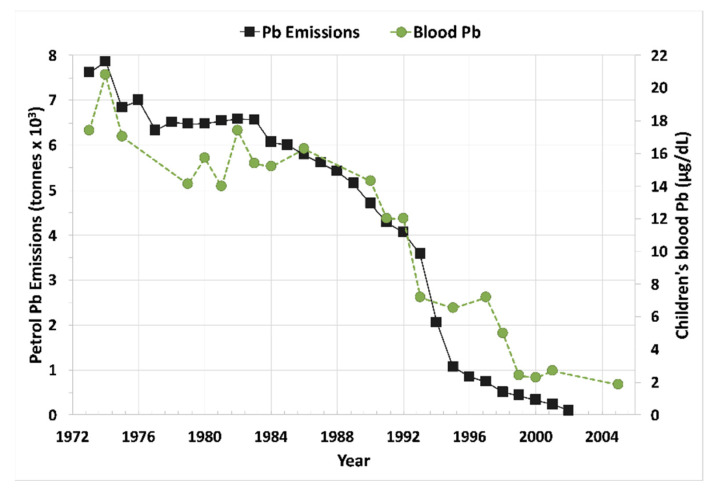
Blood Pb increases and decreases from 1973 to 2005 correspond with increases and decreases in leaded petrol emission ending during 2002 in Australia. Although the leaded petrol emissions continued to decrease, blood Pb stabilized at around 2 µg/dL. Revised and redrawn from [19].

### 3.2. Air Pb, Soil Pb, and Blood Pb

#### 3.2.1. London, UK, Evidence for 20th Century Pb Deposition being Remobilized into the Atmosphere

Britain banned leaded petrol at the end of the 20th century [26].Prior to the ban, Pb from exhaust particles were deposited in soils.Leaded petrol was the main source of Pb during the 20th century.Pb isotopes of air particles collected in London (2014–2018) were measured [26].Pb isotopic composition of air particles matched road dust and topsoil Pb.Persistence of Pb indicates that London reached an air Pb baseline.The policy measures in London are insufficient to decrease the Pb baseline [26].

#### 3.2.2. New Orleans, LA, USA, the Concurrent Decrease of Soil Pb and Blood Pb

A 19-year study evaluated changes in soil Pb and blood Pb after the 1986 rapid phasedown of leaded gasoline [27].Reservoirs of soil Pb persist in topsoil from fuel emissions and other Pb sources, which can become Pb dust.Over ~15 years, the median soil Pb decreased from 99 to 54 mg/kg as mapped in Figure 6. The spatiotemporal declines in soil Pb and blood Pb are presented in Figure 7.
Figure 6Spatiotemporal declines of soil Pb in New Orleans. The maps of New Orleans were created from a citywide survey of the metropolitan communities (*n*~5400 soil samples systematically collected across 274 census tracts) [27].
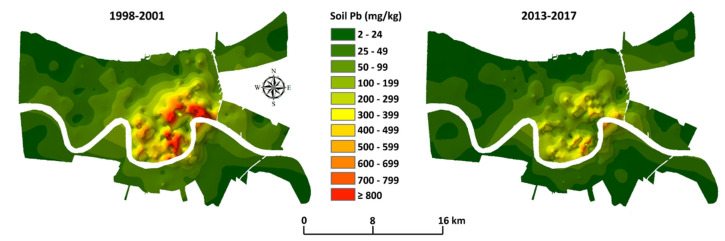

Figure 7Graph of the median blood Pb (µg/dL) on the y-axis and median soil Pb (mg/kg) on the x-axis. Two surveys of New Orleans over an interval of ~15 years illustrate the concurrent decrease in both blood Pb and soil Pb. The high blood Pb census tracts correspond with high soil Pb. The Fisher’s Exact *p*-values are <10^−36^. As illustrated in Figure 5, the most contaminated areas are clustered in the inner-city where the highest children’s blood lead values are found. Combination of two different figures and redrawn [27].
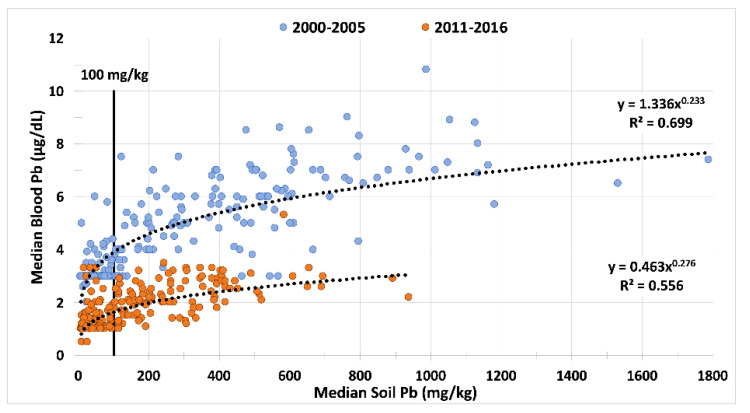
Over ~12 years, median blood Pb declined from 3.6 µg/dL to 1.2 µg/dL.Fisher’s exact test result is <10^−36^ and the null hypothesis is rejected.Similar declines are hypothesized in other major cities. The results of a follow-up study in the Detroit Tri County Area of Michigan helps validate the hypothesis [28].

## 4. Discussion

### 4.1. Ongoing Pb Exposure: Exogenous Soil Pb, Air Pb, and Endogenous Sources

Overall, these studies from around the world show that leaded gasoline emissions created high levels of air Pb, which resulted in high levels of children’s blood Pb, as well as high levels of soil Pb. Results from the UK, New Zealand, Mexico, Durban SA, Korea, NW Europe, and the US show that the decline in leaded petrol air emissions was also a driver of reductions in blood Pb levels. US agencies generally focused on lead-based paint and associated Pb dust as the primary source of lead poisoning and followed a “soil Pb is due to old leaded paint” mantra promoted by the lead industry [7]. Most European countries agreed to ban the use of lead-based paint for household use in the 1920s; the US did not agree to the ban until 1978. Pb in household paint was then regulated to 600 µg/g, still a significant amount of Pb [29]. While leaded paint is undeniably a local exposure source, we call attention to the rapid decline in blood Pb following the cessation of leaded petrol as shown by Figure 2, Figure 3, Figure 4 and Figure 5 and backed by U.S. blood Pb reports [30,31], demonstrating the importance of atmospheric Pb from leaded petrol as a major source of exposure. However, despite the sharp decreases in air Pb, the blood Pb responses were not as steep. Furthermore, as illustrated globally by Figure 2, Figure 3, Figure 4 and Figure 5, population blood Pb stabilizes at concentrations above zero, levels which clinicians regard as unsafe [5,29].

Studies conducted in Korea and NW Europe provide insights into the causes of the continued above zero µg/dL Pb exposure of the population. Korea observed that further regulation of other sources of Pb such as industrial, food, and baby products are required to reduce exposure [24]. Studies in NW Europe and the discrepancy between air Pb and blood Pb suggests alternative Pb sources continue to present exposure. These include exogenous (outside the body) and endogenous (inside the body) Pb sources [25].

Exogenous sources of Pb include inhaled air Pb [1]. Even though air Pb has declined, it persists as a legacy source of Pb. As described in Section 3.2.1, the findings in London show that current atmospheric Pb isotopes match the soil and dust Pb, originally emitted to the environment from vehicles using leaded petrol [26]. Studies from New Orleans, described in Section 3.2.2, illustrate that Pb contaminated soil is strongly associated with blood Pb [27]. The atmospheric Pb becomes settled Pb dust in the environment. Lead dust in soil is a reservoir of Pb and consequently, a source for further Pb exposure via inhalation and dust ingestion. As demonstrated in London, air Pb, driven by remobilization from the past accumulation of soil Pb, appears to have become an important contributor to the Pb baseline in urban environments. The amount of soil Pb depends on the size of an urban area; the larger the city, the more highly contaminated the soils will be [32]. Here, we place the research from London into global context, connecting the available data on air Pb, soil Pb, and blood Pb, to show how ubiquitous and harmful these legacy Pb sources are.

Endogenous sources include Pb in human tissues and particularly the stores in bones. Calcium (along with Pb) stored in the bones is released to maintain homeostasis of metabolic Ca and is part of the turnover of bone tissue [33]. Diet impacts blood Pb and urine in adult women with consequences on the mobilization of Pb from bone stores [34]. Children are especially vulnerable to Pb exposure because of their Ca requirements. Adult cognitive functions are more strongly associated with biomarkers of cumulative dose (mainly lead in tibia) than with blood lead levels [35]. The exogenous Pb pathway results in the transfer of Pb to the blood stream and then to the bones where Pb is endogenously stored. If contact with Pb is curtailed, then the amount of endogenous Pb in bone will also decline.

These data demonstrate that endogenous sources of Pb arise from prior and ongoing exogenous Pb exposure. As other research has shown, the most effective way to reduce blood Pb is to limit further exogenous Pb exposure. As Clair C. Patterson asserted to the National Academy of Sciences in 1980, “Sometime in the near future it probably will be shown that the older urban areas of the United States have been rendered more or less uninhabitable by the millions of tons of poisonous industrial lead residues that have accumulated in cities during the past century” [36]. Reducing leaded gasoline worldwide was a tremendous step towards maintaining the habitability of cities, but further efforts are needed to reduce exposure to the reservoir of soil Pb that can be remobilized to the atmosphere.

### 4.2. New Approaches for Primary Prevention

Here, we show data from around the world indicating that soil Pb plays a critical role in human Pb exposure. Collectively, these data demonstrate the urgent need to prevent remobilization of environmental Pb to the atmosphere [26]. Reducing human Pb exposure and advancing primary prevention thus requires novel polices to decrease exogenous contact to the reservoir of Pb in soil and curtailing remobilization of soil Pb into the atmosphere [26]. A major precedence for primary prevention of children’s contact with polluted soil was taken by Norway with a national action plan for mapping and remediating soils at childcare centers and playgrounds [37]. Numerous case studies and evidence-based approaches for addressing Pb-contaminated urban soil have been shown to be effective, namely by removing or simply covering contaminated soils with new clean soils [38]. Addressing soil contamination and changing vacant land into green spaces has also resulted in improvements in community mental health in Philadelphia [39]. New York City has created a Clean Soil Bank, which transports excavated subsoils from construction projects for beneficial use to reduce soil Pb exposure and promote community gardening and urban agriculture using low Pb soils [40].

Towards the goal of addressing the soil contamination issue, soil Pb must be measured and mapped to identify priority areas for primary prevention. Foundational work has been carried out towards this end: Markus and McBratney (2001) summarized spatial Pb distribution in soils from countries and cities around the world [41], and Datko-Williams et al. (2014) reviewed soil Pb concentrations across 62 cities in 29 US states between 1970 and 2012 [42]. Soil Pb mapping has occurred in Baltimore, MD [43], New Orleans, LA [25], Minneapolis, MN [44], Oakland, CA [45], New York City, NY, [46], Los Angeles, CA [47], Syracuse, NY [48], Greensboro, NC [49], Torino, Italy [50], London, UK [51], and nationwide in China [52]. Walls et al., (2022) analyzed a network of community-engaged researchers working to mitigate exposure to soil Pb [53]. Numerous engaged-scientists have co-created research with impacted communities to address soil Pb in Indianapolis, IN [54], Dewey-Humboldt, AZ [55], Milwaukee, WI [56], and several Australian cities [57].

While mapping and community-centered research is foundational, primary prevention must be achieved by removing or covering contaminated soils with clean soils. As described above, there is precedence for establishing a national clean soil program in Norway that involves mapping and renovating areas of soil contamination to protect children where they play [37]. Municipalities throughout the world must undertake similar measures to decrease baseline Pb exposure. As illustrated in Figure 6 for New Orleans, cities generally have low Pb soils in their outlying areas. The combined efforts of citywide soil Pb mapping, soil construction, and soil emplacement can reduce the urban environmental Pb baseline. In turn, this can limit the remobilization of Pb dust and reduce the baseline of air Pb and blood Pb levels of populations living in contaminated communities.

## 5. Conclusions

The toxic and potentially lethal health impacts of Pb are well-known, and while great advances in primary prevention have been made, we reviewed available data from around the world that show that baseline population-level exposures are plateauing above 0 ug/dL and are still too high. Available data demonstrate that air Pb, largely from leaded petrol, has been a major exposure source over time. While leaded petrol has been banned worldwide, it has left a legacy of lead in soil, which can be remobilized to the atmosphere. We review these data together, which strongly indicate that remobilized soil Pb and dust are contributing to the elevated blood Pb of urban populations worldwide. Exogenous soil and air Pb can be stored in bones and serve as endogenous sources of exposure. Despite the attempts of industries to deny the implications of leaded petrol on health, this globally pervasive source of toxicity can no longer be ignored. Simple steps can be taken to address this issue: primary prevention can occur from removing or covering contaminated soils with clean soils. Norway has a viable example of a national action plan to map and mitigate exposure to soils at public play areas, such as parks, elementary schools, and day care centers, and New York City has created a Clean Soil Bank to provide a source of clean materials for such purposes. New policies are feasible, and needed, to facilitate mapping and mitigation of exposure to Pb contaminated and unhealthy soils. Healthy soils are essential for ensuring urban health and sustainability.

## Figures and Tables

**Figure 1 ijerph-19-09500-f001:**
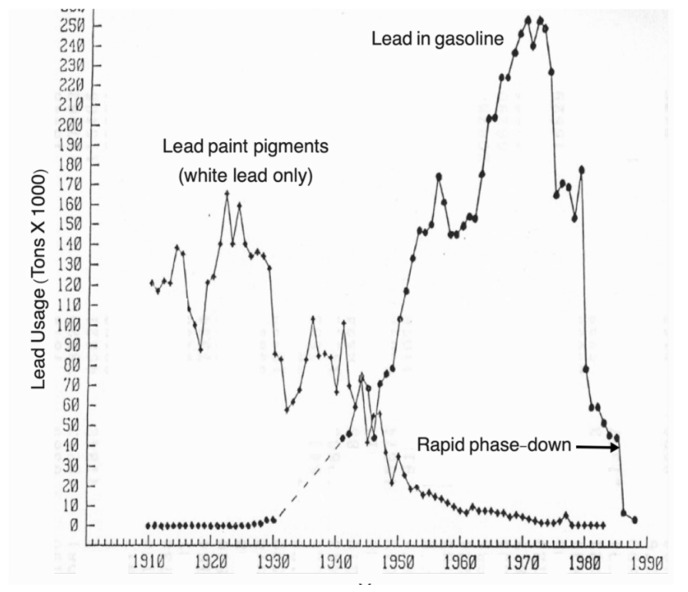
Graph comparing the U.S. annual quantities of Pb in paint and gasoline. In the late 1960s and early 1970s, appropriately sensitive analytical instruments became available to measure lead. Failure to include either one or another of the major sources is a disservice to public health. For example, Dr. Henderson’s concern about inhalation is not addressed when lead aerosols from the combustion of leaded petrol are ignored as a major source of lead.

**Table 1 ijerph-19-09500-t001:** Search results of selected countries obtained for air Pb and blood Pb.

Google Search Key Words	Publication with Short Title	Year	Ref.
Sweden, blood Pb, air Pb	Blood Pb of Swedish children, 1978–2007	2008	[15]
China, blood Pb, air Pb	Blood Pb urban and suburban children	2018	[16]
Germany, blood Pb, air Pb	Long-term trend Pb exposure adults	2021	[17,18]
Australia, leaded gas, blood	Rise, fall, and remobilization of Pb	2017	[19]
UK airborne Pb and blood Pb	Effect of Airborne Pb on Blood Lead	1983	[20]
NZ petrol Pb, blood Pb,	Petrol Pb effect on blood Pb	1986	[21]
Mexico, blood Pb and air Pb	Lead in air and traditional lead-ceramic	2018	[22]
Durban, air Pb, blood Pb	Manganese and Pb in children’s blood	2010	[23]
Korea, blood Pb, petrol Pb	Blood Pb before and after leaded petrol	2017	[24]
NW Europe, blood Pb, air Pb	Air Pb and blood Pb decline in Europe	2015	[25]

## Data Availability

Most of the articles listed include extra resources in the references for further study.

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
