# Peer review of "Lead in Air, Soil, and Blood: Pb Poisoning in a Changing World"

_ijerph, 2022, doi:10.3390/ijerph19159500_

Round 1

Reviewer 1 Report

The manuscript presented by Mielke et al. reviews the blood Pb levels and air Pb. The manuscript's main idea is very interesting and necessary to demonstrate that ban Pb use is pivotal to decreasing blood lead levels. However, the manuscript organization is not good; I could not identify if it is a systematic review or a narrative review. I would suggest organizing the paper as a narrative review without the materials and methods and results section. If the authors decide to keep the manuscript the way it is, I suggest doing a systematic search in at least other two databases and presenting in the paper the keywords and Boolean terms used in the search to make the results reproducible, as well as following the PRISMA guideline, in my opinion, is mandatory.

Author Response

The manuscript presented by Mielke et al. reviews the blood Pb levels and air Pb. The manuscript's main idea is very interesting and necessary to demonstrate that ban Pb use is pivotal to decreasing blood lead levels. However, the manuscript organization is not good; I could not identify if it is a systematic review or a narrative review. I would suggest organizing the paper as a narrative review without the materials and methods and results section. If the authors decide to keep the manuscript the way it is, I suggest doing a systematic search in at least other two databases and presenting in the paper the keywords and Boolean terms used in the search to make the results reproducible, as well as following the PRISMA guideline, in my opinion, is mandatory.

  • Thank you expressing the interesting nature of this work and the significance of demonstrating that the ban in petrol Pb use has decreased blood lead levels.
  • We agree with your suggestion that we should more carefully describe the type of review we conducted. We cite Snyder 2019, in describing our review as “integrative:” “We conducted an integrative review to provide an overview of the knowledge base and re-conceptualize the links between air Pb and blood Pb [Snyder 2019].” This was neither systematic nor narrative, but rather, a series of searches focused on a well-understood topic with a new lens.
  • We agree that adding additional searches was necessary and have included searches in two additional databases: Scopus and Web of Science. The most relevant articles are still included in Table 1 and the figures we presented. However, these additional searches provided several additional articles that we cited in the discussion.
  • In the methods section, we described our use of the Boolean terms “or” for connecting similar terms (i.e., air Pb or atmospheric lead), as well as “and” in linking each of our searches together.
  • We appreciate your reference to the PRISMA guidelines. Since we did not conduct a systematic review, we did not cite these guidelines here.

Reviewer 2 Report

The manuscript entitled with "Lead in air, soil, & blood: Pb poisoning in a changing world" aimed to provide a review on long-term Pb exposure of human subjects in relation with atmospheric Pb and settled Pb dust. The authors failed to provide any finding with scientific significance or novelty. The database for literature review is too limited. The article sumarized several related reports but is lack of critical analysis and deep discussion. Besides, the writing style also needs to be improved if as a academic article. 

Author Response

The manuscript entitled with "Lead in air, soil, & blood: Pb poisoning in a changing world" aimed to provide a review on long-term Pb exposure of human subjects in relation with atmospheric Pb and settled Pb dust. The authors failed to provide any finding with scientific significance or novelty. The database for literature review is too limited. The article sumarized several related reports but is lack of critical analysis and deep discussion. Besides, the writing style also needs to be improved if as a academic article. 

  • Thank you for your consideration of this manuscript.
  • We agree that the literature review was limited and have thus included additional searches in two databases: Scopus and Web of Science. In the Methods section, we have included additional description of our review as integrative, as opposed to systematic or narrative. We have also included the key words and Boolean terms we used.
  • We have worked to revise the manuscript to highlight our critical analysis of the underexplored links between leaded gasoline, air lead, and blood lead. We have revised to include thorough discussion of these links and steps for mitigation.
  • While we did not conduct original research for this report, we contend that this review and synthesis contributes to the literature by highlighting a globally pervasive and deeply impactful public health issue in need of immediate attention.

Reviewer 3 Report

This paper is titled “Lead in air, soil, & blood: Pb poisoning in a changing world”.
This is a meaningful topic, but the content of the article still needs to be further improved to make the paper better.
Therefore, I have made some comments in order to improve the presentation of the findings.

1Line13: This sentence is more like part of the method than background.

2I suggest that the introduction part needs to be rethought.This study should clearly state what the current research is flawed and what ideas this study can provide to address this issue.

3Line47-48:The first is the language problem,curtailing & decreasing. Besides,the article may draw attention to lead poisoning and provide decision-making basis for human lead poisoning protection, but it cannot reduce the role of atmospheric lead and deposited lead dust in human lead poisoning.

4Line84: This result contradicts the overall conclusion, and the authors seem to have no explanation.

6Line92-94: I can't get the relevant results from the figure or table in the article.

7Line109: From the figure, the lead emission data ends in 2002, why is the text description 1973-2015?

8Line150-169: This paragraph is rather confusing. A lot of words were used to describe the problem of lead paint before, but it was finally involved that leaded gasoline was the main source of lead in the atmosphere.

9Line207-209: There does not seem to be such a logical relationship between these two sentences.

10I suggest that the article should have a complete structure, and the conclusions of the research need to be presented.

Author Response

This paper is titled “Lead in air, soil, & blood: Pb poisoning in a changing world”.
This is a meaningful topic, but the content of the article still needs to be further improved to make the paper better.
Therefore, I have made some comments in order to improve the presentation of the findings.

1、Line13: This sentence is more like part of the method than background.

Thank you for this suggestion. We have moved this statement to Methods and have revised the abstract to clearly delineate each section (Background, Methods, etc.).

2、I suggest that the introduction part needs to be rethought. This study should clearly state what the current research is flawed and what ideas this study can provide to address this issue.

Thank you for this helpful suggestion. We have rephrased the intro to emphasize the lack of recognition of the impacts of leaded gasoline on soil lead, air lead, and blood lead.

3、Line47-48: The first is the language problem, curtailing & decreasing. Besides,the article may draw attention to lead poisoning and provide decision-making basis for human lead poisoning protection, but it cannot reduce the role of atmospheric lead and deposited lead dust in human lead poisoning.

Thank you for pointing this out. We have rephrased this statement.

4、Line84: This result contradicts the overall conclusion, and the authors seem to have no explanation.

Thank you for this comment. We have clarified that blood Pb data are from urban surveys, while atmospheric Pb data also include industrial areas.  

6、Line92-94: I can't get the relevant results from the figure or table in the article.

Thank you for this comment. We have removed this sentence since it is not related to the overall goals of the manuscript.

7、Line109: From the figure, the lead emission data ends in 2002, why is the text description 1973-2015?

Thank you for pointing this out. We have adjusted the caption to reflect that the data end in 2005.

8、Line150-169: This paragraph is rather confusing. A lot of words were used to describe the problem of lead paint before, but it was finally involved that leaded gasoline was the main source of lead in the atmosphere.

Thank you for highlighting this confusion. We have removed the section on leaded gasoline, placing part of this in the introduction and part in the first section of the discussion.

9、Line207-209: There does not seem to be such a logical relationship between these two sentences.

Thank you for addressing this issue. We have included a direct quotation from Dr. Clair C. Patterson on the potential inhabitability of cities to link these two topics.

10、I suggest that the article should have a complete structure, and the conclusions of the research need to be presented.

Thank you for this suggestion. We have now included a conclusion section.

Round 2

Reviewer 1 Report

No comments.

Reviewer 2 Report

After a major revision, the manuscript was improved and the problems I previously concerned have been fixed. Overall, this manuscript meets the interests and the quality requirement of this journal. I think it can be accepted in the current form.

Reviewer 3 Report

According to the modification, it is agreed to accept the article.

This manuscript is a resubmission of an earlier submission. The following is a list of the peer review reports and author responses from that submission.

Round 1

Reviewer 1 Report

The revised MS has supplemented much case studies, and could be considered for acceptance.

Reviewer 2 Report

Regarding the manuscript entitled “Air Pb, soil Pb, & blood Pb: Pb poisoning in a changing world”. The manuscript deals with an interesting topic and it is well-written. I am not sure if the work was submitted as a review article, but in my opinion, this is clearly what it is. The bibliography that is used in this review article is very limited (and old in some cases). Section 3 (which should not be named results), includes data from only 6 publications, which are presented in a very basic way. Pb as part of the aerosols, which is by far the most important exposure pathway, is barely discussed and relevant studies are not included.  Taking everything into account, I don’t find the study suitable for publication.

First of all the authors need to clarify if this is an article or a review. I was not sure how to judge the article if it is not clear what it is. The work is submitted as an article, but only presents data from previous publications. If it is indeed an article, then it should be rejected, no questions asked, because it does not include any experimental work at all.

If it is a review, then sections 3 and 4 should be merged and more studies need to be included. In my opinion, the keywords the authors used are too few, and that leads to excluding a large number of publications that describe the effect of Pb as a component of aerosols. I did not make specific comments, because, again in my opinion, there are fundamental issues with the manuscript and the details are not important on this point. 

Reviewer 3 Report

The document discusses Pb - air, soil, and blood. It lists the results of a Google Scholar search but does not describe keywords or provide a comprehensive list. It is also not well-structured or described.

Line 6: There is no 3rd author affiliation.

Line 15: The document looks at multiple locations so it's unclear why only New Orleans data is mentioned.

The Abstract does not need the sections to be numbered (i.e., "(1)", etc.).

The Introduction is extremely short. The authors should consider elaborating the important points they bring up.

Table 1: It is unclear if this is the complete set of search results or an example of search results.

The Methods section does not describe the specific words used for the Google Scholar search. For a review, this must be clearly described.

Line 66: These studies were not listed in either Table 1 or as keywords in the Methods section.

Line 103: Should this be "1980s" instead of "1880s"?

Line 144: Subheading 4.1 should have a title (same comment for subsequent subheadings).

Lines 210-11: Section 4.4 should not necessarily describe 4.5 especially as it is the next sentence.

There does not seem to be a clear conclusion.